# *MdGRF22*, a 14-3-3 Family Gene in Apple, Negatively Regulates Drought Tolerance via Modulation of Antioxidant Activity and Interaction with *MdSK*

**DOI:** 10.3390/plants14131968

**Published:** 2025-06-27

**Authors:** Jiaxuan Ren, Hong Wang, Mingxin Zhao, Guoping Liang, Shixiong Lu, Juan Mao

**Affiliations:** 1Institute of Forestry, Fruits and Floriculture, Gansu Academy of Agricultural Sciences, Lanzhou 730070, China; rjx6642@163.com (J.R.); wanghong@gsagr.ac.cn (H.W.); zmx850312@163.com (M.Z.); 2College of Horticulture, Gansu Agricultural University, Lanzhou 730070, China; xscjy1gsau@163.com (G.L.); 18893912407@163.com (S.L.)

**Keywords:** apple, *MdGRF22*, Y2H, antioxidant system

## Abstract

The 14-3-3 proteins play crucial roles in regulating plant growth, development, signal transduction and abiotic stress responses. However, there exists a scarcity of research on the role of 14-3-3 proteins in responding to abiotic stress in apples. In this study, we isolated the *MdGRF22* gene from the apple 14-3-3 family. Through the screening of interacting proteins and genetic transformation of *Arabidopsis thaliana* and apple callus tissues, the function of the *MdGRF22* gene under drought stress was verified. The coding sequence (CDS) of *MdGRF22* consists of 786 bp and encodes for 261 amino acids. Through sequence alignment, the conserved 14-3-3 domain was identified in *MdGRF22* and its homologous genes, which also share similar gene structures and conserved motifs. Subcellular localization revealed that the MdGRF22 protein was predominantly located in the cytoplasm and cell membrane. The yeast two-hybrid (Y2H) analysis demonstrated a possible interaction between *MdGRF22* and *MdSK*. In addition, *MdGRF22* transgenic plants generally exhibited lower superoxide dismutase (SOD), catalase (CAT) and peroxidase (POD) activities, higher malondialdehyde (MDA) levels and relative electrolyte leakage under drought conditions compared with wild-type (WT) plants. Our study suggests that *MdGRF22* may reduce the drought resistance of transgenic *A. thaliana* and callus tissues by interacting with *MdSK.* This study provides a theoretical basis for further exploring the function of 14-3-3 family genes.

## 1. Introduction

*Malus domestica* Borkh., a deciduous Rosaceae species of significant nutritional and economic importance, is cultivated globally. Nevertheless, drought persists as a major constraint on apple production. Prolonged aridity reduces soil water retention, consequently inhibiting tree growth and manifesting as foliar chlorosis, premature abscission of leaves, and reduced fruit set—ultimately diminishing the yield [1]. Additionally, drought causes slow growth and the development of branches, smaller fruits, and severely affects the quality and fruit setting rate [2]. Moreover, high temperatures are not conducive to fruit coloring, reducing the marketability of the fruit. Therefore, research on enhancing the drought resistance of apple varieties is particularly important. Apples, due to their long juvenile period and highly heterozygous genotype, exhibit trait segregation in their hybrid offspring. As a result, multiple excellent traits cannot coexist in a single apple variety. Genetic engineering is often used to purposefully improve the genetic traits of fruit trees, and it is of great significance for gene exploration and the cultivation of stress-resistant fruit trees [3]. Protein 14-3-3, a stress-inducing protein, plays an important role in plants’ response to abiotic stress.

In 1967, Moore and Perez isolated a protein from the bovine brain and named it 14-3-3 due to its elution and migration pattern with diethylaminoethyl (DEAE) cellulose chromatography and starch-gel electrophoresis [4]. The 14-3-3 protein is conserved in plants, with the N-terminal region critical for binding with different types of membranes, and the C-terminal region serving as the site of protein–protein interactions [5]. The 14-3-3 proteins are ubiquitous in eukaryotic cells, predominantly in the cytoplasm, but are also found in the plasma membrane, nucleus, chloroplast and mitochondria [6]. For example, the *GRF2* gene encoding a 14-3-3 protein subtype in *Arabidopsis thaliana* is distributed among multiple organelles, with its primary role being in the cytoplasm [7]. The 14-3-3 protein lacks enzymatic catalytic activity and primarily performs its regulatory function through binding to target proteins in response to various regulatory mechanisms. Moreover, 14-3-3 proteins can specifically recognize phosphorylated target proteins, thereby preventing dephosphorylation and modulating a variety of biological processes by altering the activity of target proteins [8].

Drought environments lead to insufficient water content in plants, reducing their resilience, and negatively impacting physiological metabolism, growth and development, which often results in plant death [9]. Studies have shown that 14-3-3 proteins act as stress-responsive proteins under abiotic stress. However, previous studies have drawn different conclusions about the effects of 14-3-3 protein on drought tolerance. On the one hand, 14-3-3 proteins significantly enhance drought tolerance in plants. *OsGF14c*, a subtype of the rice GF14 family, has been identified as a positive regulator of drought tolerance [10]. On the other hand, 14-3-3 proteins have also been found to negatively impact drought tolerance in plants. The overexpression of soybean *GsGF14ο* could reduce the drought tolerance of *A. thaliana* by regulating stoma size [11]. The role of 14-3-3 proteins in the drought resistance of apples has been studied. Ren et al. [12] found that the apple *MdGRF11* gene mainly enhances the response to drought stress by regulating the rate of water loss and the opening of stomata, as well as by up-regulating antioxidant and stress-related genes. Furthermore, *MdGRF11* accumulates responses to drought at the protein level, and it removes reactive oxygen species by influencing the expression of downstream genes of *MdARF19-2*, thereby enhancing the plant’s resistance to drought stress [13]

The plant glycogen synthase kinase 3 (GSK3) protein has evolved some complex mechanisms to respond to abiotic stress. Glycogen synthase kinase GSK3, also known as the shaggy-like kinase (SK), mainly functions as a glycogen synthase [14,15]. Previous studies have shown that the GSK3 protein is involved in the response of plants to drought stress. Functionally conserved across species, these kinases act as critical signaling hubs that coordinate drought responses. For instance, in *A. thaliana*, subgroup II kinase BIN2 phosphorylates the NAC transcription factor RD26, thereby enhancing its transcriptional activation of drought-responsive genes and promoting tolerance [16]. Similarly, BIN2-mediated phosphorylation stabilizes the AP2/ERF factor TINY, linking brassinosteroid signaling with drought response suppression under non-stress conditions [17]. Post-translational activation of GSK3 kinases by drought signals facilitates substrate modifications. Stress perception was observed to promote BIN2-RD26 physical interaction, enabling site-specific RD26 phosphorylation that amplifies drought gene expression [16]. This modification alters substrate stability and subcellular localization—notably inducing the nuclear retention of TINY—to integrate stress signaling [17]. Collectively, these evolutionarily conserved pathways position GSK3 kinases as prime targets for engineering drought-resilient crops through transcriptional network modulation

14-3-3 is a highly conserved acidic protein that is ubiquitous in plants. Although the roles of 14-3-3 family genes have been explored in numerous species, the function of 14-3-3 family genes under abiotic stress is still relatively unknown in apple. This study aims to elucidate the role of *MdGRF22*, a 14-3-3 family gene in apple, in drought stress response. Specifically, we investigated its structural features, expression pattern, subcellular localization, potential interacting proteins, and functional validation via transgenic *A. thaliana* and apple callus assays. By analyzing antioxidant enzyme activities and physiological indices under drought stress, we provide insights into *MdGRF22* as a negative regulator of drought tolerance, offering a foundation for improving stress resilience in apple cultivars.

## 2. Results

### 2.1. Sequence Alignment of MdGRF22 and Homologous Genes

To characterize the sequence and structure of *MdGRF22*, we performed an analysis of the protein sequence of the *MdGRF22* gene. The sequence analysis of the *MdGRF22* gene revealed that the protein encoded 261 amino acids (Appendix A). The sequence alignment showed significant sequence similarity and a 14-3-3 conserved domain between *MdGRF22* and its homologous gene (Figure 1A), with the protein structure of *MdGRF22* predominantly consisting of alpha helices (Figure 1B). A phylogenetic tree was constructed using nine 14-3-3 protein sequences, and MdGRF22 is genetically most closely related to XM_009347515.3 (*Pyrus bretschneideri*) (Figure 1C), which is presumed to share a similar function with it. Additionally, *MdGRF22* and its homologous genes exhibit similar gene structures and motif distributions (Appendix A).

### 2.2. Quantitative Real-Time PCR (qRT-PCR) and Subcellular Localization Analysis of MdGRF22 Gene in Apple

We analyzed the expression pattern and subcellular localization of the *MdGRF22* gene under drought stress. The qRT-PCR analysis of ‘Royal Gala’ apple plantlets at different times revealed that the expression level of *MdGRF22* was down-regulated with the increase in time under PEG treatment (Figure 2A). Meanwhile, tissue expression indicated that *MdGRF22* was predominantly expressed in stems [18]. Thus, we hypothesize that the *MdGRF22* gene might negatively regulate the response to drought stress. The cDNA of ‘Royal Gala’ apples was amplified via PCR, and the resulting bands are depicted in Appendix A. The recombinant plasmid was transformed into E. coli DH5α, and PCR detection confirmed the presence of the expected 786 bp sequence (Appendix A). Subsequently, the product was introduced into Agrobacterium, and the PCR analysis revealed a suitable band (Appendix A).

Using the pCAMBIA1300-GFP empty vector as a positive control, the *MdGRF22* recombinant plasmid was expressed instantaneously in tobacco leaves. The results indicate that the fluorescence localization of the pCAMBIA1300-*MdGRF22*-GFP expression vector is in the cytoplasm and cell membrane (Figure 2B).

### 2.3. Y2H Analysis of MdGRF22 Protein

To identify the potential interacting proteins of MdGRF22, we conducted a screening using Y2H. The recombinant plasmid MdGRF22-pGBKT7 (BD) was transfected into the yeast strain AH109, and white spots were observed on both SD/-Trp and SD/-Trp + 5-bromo-4-chloro-3-indolyl-α-D-pyrangalactoside (X-α-gal) plates after two days. These observations suggest that the *MdGRF22* gene did not exhibit self-activation (Appendix A).

Using MdGRF22-BD as a bait protein, 87 monoclonal strains were obtained on QDO (SD/-Ade/-His/-Leu/-Trp/) plates by nuclear double hybridization. A total of 32 strains either exhibited spots or turned blue on the QDO + X-α-gal plates (Appendix A). Twenty distinct products were identified by PCR from these strains, with a total of six bands of different sizes (Appendix A). The majority of identified proteins were determined to be associated with abiotic stresses through NCBI alignment, including SK (shaggy-like kinase), NADH1 (enoyl-[acyl-carrier-protein] reductase NADH 1), EF1 (elongation factor 1-alpha) and DNAJ6 (chaperone protein dnaJ A6).

The MdGRF22-BD and MdSK-pGADT7 (AD) recombinant plasmids were co-transformed into the yeast strain AH109 (Appendix A). BD/AD, MdSK-AD/BD and MdGRF22-BD/AD were applied to DDO (SD/-Leu/-Trp), DDO + X-α-gal, QDO and QDO + X-α-gal plates, respectively. Following the co-transformation of MdGRF22-BD with MdSK-AD, blue spots appeared on DDO + X-α-gal and QDO + X-α-gal plates, indicating that there may be an interaction between MdGRF22 and MdSK proteins (Figure 3).

### 2.4. Overexpression of MdGRF22 Reduced Drought Tolerance in Transgenic A. thaliana

To validate the role of *MdGRF22* in drought tolerance, we generated genetically transformed *A. thaliana* plants. Transgenic *A. thaliana* seedlings were screened in Hyg-containing medium through T3 generation (Figure 4A). DNA was extracted from the leaves of T3 *A. thaliana* plants, using the recombinant plasmid *MdGRF22*-pCAMBIA1300 as a positive control and the WT as a negative control for the identification of transgenic strains. PCR amplification results revealed that lines OE-1, OE-2 and OE-3 exhibited bands matching the size of the positive control, whereas no bands were observed for the WT (Figure 4B), indicating the successful transformation of the *MdGRF22* gene into *A. thaliana*.

Two-week-old WT and transgenic *A. thaliana* lines OE-1, OE-2 and OE-3 were subjected to natural drought conditions for 15 days, with the transgenic lines exhibiting varying levels of wilting and the WT demonstrating more robust growth. Following a two-day recovery period, the transgenic lines partially resumed growth, whereas the WT continued to grow normally (Figure 4C). Physiological indices indicated that the relative electrolyte leakage and MDA content in leaves of *MdGRF22* transgenic *A. thaliana* were substantially higher compared to the WT under drought conditions, with the MDA content in line OE-2 reaching 114.71 nmol·g^−1^ (Figure 4D,E). Furthermore, the activities of CAT, SOD and POD in the transgenic lines increased. However, they remained lower than those in the WT, with no significant differences observed among lines OE-1, OE-2 and OE-3 after drought treatment (Figure 4F–H). These results suggest that overexpression of *MdGRF22* reduced the drought tolerance of transgenic *A. thaliana*.

### 2.5. Overexpression of MdGRF22 Reduced Drought Tolerance in Transgenic Apple Callus

We investigated the role of the *MdGRF22* gene under drought stress by transforming apple callus. Through genetic transformation and subsequent plate screening, a robust apple callus was obtained (Figure 5A). Subsequently, the callus cultured on the resistant culture medium was confirmed by PCR, where the fragment size matched that of the positive control, signifying the successful transformation of the transgenic callus (Figure 5B).

We treated WT and transgenic callus OE-1 and OE-3 with the addition of 3% and 6% PEG. Following a 15-day treatment, no significant differences in growth and fresh weight were observed between the WT and transgenic callus treated with 3% PEG (Figure 5C,D). However, under 6% PEG treatment, the fresh weight of the transgenic callus was markedly reduced compared to the WT, with weaker growth observed. Specifically, the fresh weight of OE-3 (0.122 g) was the lowest and was further diminished under 6% PEG treatment compared to the 3% PEG treatment.

Under control conditions, no differences were observed in the levels of MDA, POD, SOD and CAT between the WT and transgenic callus. However, the MDA content in the transgenic callus subjected to 6% PEG treatment exceeded that of the WT, with OE-3 exhibiting the highest MDA content at 18.65 μmol·g^−1^ (Figure 5E). No significant differences were noted in MDA, POD, SOD and CAT activities between the WT and transgenic callus under 3% PEG treatment. Conversely, under 6% PEG treatment, the activities of MDA, POD, SOD and CAT in the transgenic callus were considerably reduced compared to those in the WT (Figure 5E–H). These results indicate that overexpression of *MdGRF22* decreased the drought resistance of transgenic apple callus by decreasing the activity of antioxidant enzymes.

## 3. Discussion

Studying the functions of plant genes is helpful for understanding their growth, development and adaptive biological processes. Protein 14-3-3 is a type of acidic protein that is widely present in plants and other eukaryotic organisms [19]. Multiple sequence alignment analyses of *MdGRF22* and its homologs revealed extremely high sequence similarity, with similar sequences forming the conserved domain of 14-3-3. Alignments of 14-3-3 protein sequences from lower to higher plants, such as rice, *A. thaliana*, *P. bretschneideri*, *Marchantia polymorpha* and algae, revealed highly conserved amino acid sequences. Sequence similarities remained above 80%, which is consistent with the results of previous studies [5]. Phylogenetic trees can be used to analyze the evolutionary processes of genes among different species and describe the degree of their relatedness [20,21]. In this study, homologs of the *MdGRF22* gene were identified in various species through NCBI sequence alignment. XM_009347515.3 and *MdGRF22* are located adjacently on the evolutionary tree branch (Figure 1), and since both apple and white pear are part of the rose family, they may share similar functions and expression patterns, warranting further exploration.

Subcellular localization can reveal the specific location of a protein’s action or product, which is essential for its proper functionality [22]. Studies have demonstrated that 14-3-3 proteins are predominantly intracellular in function [6,23]. Ren et al. [12] observed through transient expression in tobacco that the fluorescence distribution of the *MdGRF11* protein was uniform throughout the cell. Zuo et al. [24] reported that four Md14-3-3 proteins (MdGF14a, MdGF14d, MdGF14i and MdGF14j) exhibited fluorescence in both the cytoplasm and nucleus. Additionally, Jia et al. [25] found that the tomato SlTFT4 protein was primarily localized within the cytoplasm and at the cell membrane based on transient transformation in tobacco leaves. In this study, the recombinant plasmid pCAMBIA1300-*MdGRF22*-GFP was transiently expressed in tobacco, revealing that the MdGRF22 protein was predominantly localized within the cytoplasm and at the cell membrane (Figure 2), aligning with previous research findings. It has been shown that the subcellular distribution of 14-3-3 proteins is highly dependent on their interactions with targets [26]. The binding of 14-3-3 proteins to target proteins can also alter their subcellular localization [27]. Nevertheless, the precise mechanisms underlying the interaction between MdGRF22 and its target proteins, particularly regarding the modulation of subcellular localization and gene regulation, warrant additional investigation.

The various 14-3-3 isomers can form different homologs or heterodimers, which demonstrates the diversity and specificity of their functions [28]. The 14-3-3 proteins interact with other proteins to regulate the subcellular localization, activity, conformational change, and stability of their phosphorylation targets. These interactions are also influenced by the internal and external environments of plant cells [29]. Recent research has increasingly focused on the interactions between 14-3-3 proteins and diverse binding proteins [30,31]. Our findings, utilizing Y2H analysis, indicate a potential interaction between the shaggy-like kinase MdSK and MdGRF22 (Figure 3). This interaction suggests a role for *MdGRF22* in mediating drought stress responses. SK proteins are known regulators of multiple signaling pathways and abiotic stress tolerance [32]. Overexpression of *TaSK5*, an ortholog in *A. thaliana*, significantly enhances plant tolerance to both salinity and drought [33]. Furthermore, qRT-PCR analysis revealed that *MmSK* transcript levels respond dynamically to various abiotic stresses, including drought, salt, low temperature and abscisic acid (ABA) treatment, compared to unstressed controls. Specifically, both salt and drought stress induced *MmSK* expression, with transcript abundance showing distinct temporal patterns under prolonged drought conditions [34]. The apple 14-3-3 family proteins are conserved at the N-terminal and interact with other proteins at the C-terminal to regulate protease activity and function [35,36]. The interaction between MdGRF22 and MdSK may inhibit the phosphorylation regulatory site of *MdSK* and reduce its transcriptional activation of drought response genes, thereby reducing the drought resistance of plants [37,38]. At the same time, MdGRF22 may interact with MdSK to form a complex that inhibits the ABA signal, affecting the drought resistance ability of MdGRF22 protein [39]. The binding site between MdSK and MdGRF22 is still unclear. Whether the function changes after binding and the mechanisms related to reactive oxygen species (ROS)-clearance and ABA signal need further research.

To cope with the effects of adverse environments, plants have evolved a series of mechanisms to cope with stress, ensuring normal growth and reproduction [40]. In higher plants, 14-3-3 proteins form a family of regulatory proteins that are involved in regulating plant development and stress responses [27]. SOD, POD and CAT facilitate the removal of reactive oxygen species and free radicals, playing a role in maintaining the balance of reactive oxygen species [41]. The growth of *MdGRF22* transgenic callus is less robust than that of the WT under 6% PEG treatment, and the growth of the transgenic *A. thaliana* was weaker than that of the WT under natural drought stress, with lower POD, SOD and CAT activities than the WT (Figure 4 and Figure 5). Furthermore, the overexpression of *MdGRF22* led to severe membrane lipid peroxidation and loss of cell integrity in the transgenic materials, inhibited the activity of related antioxidant enzymes and increased the accumulation of ROS in the body, indicating that the *MdGRF22* gene is a negative regulatory factor for plants under drought stress. In conclusion, we proposed an improved model, where the MdGRF22-MdSK complex may inhibit the antioxidant system, resulting in weakened SOD/CAT/POD activities, an accumulation of ROS causing cellular oxidative damage, and an increase in MDA content and electrolyte leakage (Figure 6). This is in stark contrast to the enhancement of antioxidant capacity and stress gene expression by the combination of *OsGF14c* and *ZmGF14-6* in rice [42,43], and is opposite to the functional similarity of *MdGRF22* with negative regulatory factors of 14-3-3 such as *OsGF14b* [44] and *GsGF14o* [11]. In conclusion, our research has found that *MdGRF22* may interact with *MdSK*, thereby reducing the drought resistance of transgenic materials. Although this study identified the interaction between *MdGRF22* and *MdSK* through Y2H assay, it is necessary to further validate the accuracy of the results using co-IP or BiFC techniques. Secondly, the study on the function of *MdGRF22* under abiotic stress should be further validated by combining RNAi and gene editing techniques. The specific mechanism of interaction between *MdGRF22* and *MdSK*, as well as the regulatory mechanism under stress conditions, require further investigation.

## 4. Materials and Methods 

### 4.1. Sequence and Structure Analysis of MdGRF22

We searched for homologs of the *MdGRF22* (MD10G1084500) coding sequence (CDS) in NCBI (https://www.ncbi.nlm.nih.gov/) (accessed on 9 July 2023) and performed multiple sequence alignment analyses using DNAMAN 5.2. Meanwhile, the phylogenetic tree was constructed using the neighbor joining (NJ) method with MEGA 7.0, and the gene structure and motifs were analyzed using GSDS 2.0 (http://gsds.gao-lab.org/) (accessed on 15 July 2023) and MEME 5.5.8 (https://meme-suite.org/meme/tools/meme) (accessed on 25 August 2023), respectively. The protein model of *MdGRF22* was predicted by SWISS-MODEL (https://swissmodel.expasy.org/interactive) (accessed on 18 November 2023).

### 4.2. Plant Materials and Treatment

We collected the explants of ‘Royal Gala’ apple seedlings planted in the solar greenhouse of College of Horticulture, Gansu Agricultural University. After rinsing with running water, the explants were disinfected with 75% alcohol and NaClO solution (NaClO:H_2_O = 1:2). After cleaning with sterile water, the explants were transferred to Murashige and Skoog (MS) medium (MS + 3% sucrose + 6 g·L^−1^ agar + 2 mg·L^−1^ 6-BA + 1 mg·L^−1^ NAA, PH = 5.8–6.0) for culture. These apple plantlets were cultured in a 16 h light cycle at 25 °C and an eight-hour dark cycle at 20 °C. Four-week-old plantlets were transferred to MS liquid medium containing 10% polyethylene glycol 6000 (PEG) for treatment, and the leaves were collected after 0, 2, 12 and 24 h, respectively. Three biological replicates were used for each treatment, with each replicate consisting of five plantlets (n = 15 total per time point). The leaves were frozen in liquid nitrogen and then transferred to −80 °C for storage.

### 4.3. qRT-PCR Analysis of MdGRF22 Gene

The RNA of the leaves from treated ‘Royal Gala’ plantlets were extracted using a plant RNA extraction reagent kit (Real Times (Beijing) Biotechnology Co., Ltd., Beijing, China) and reverse transcribed using a reverse transcription kit (Prime Script^RT^ reagent Kit, Perfect Real Time, TaKaRa, Kusatsu, Japan) into cDNA. The primers were designed by Biotech Co., Ltd. (Sangon, Shanghai, China), and the reference gene was *MdGAPDH* (Appendix A). The expression level of the *MdGRF22* gene was analyzed using the LightCycler^®^ 96 Real-Time PCR System, Roche, Basel, Switzerland. The reaction system is as follows: 1 μL of upstream primer, 1 μL of downstream primer, 6 μL of ddH_2_O, 2 μL of cDNA and 10 μL of SYBR enzyme, totaling 20 μL. The gene expression level was calculated using the 2^−∆∆CT^ method [45].

### 4.4. Cloning and Transformation

RNAPlant extraction reagent (Beijing Real-time Biotechnology Co., Ltd., Beijing, China) was used to extract RNA from ‘Royal Gala’ apple plantlets. The primers for cloning the *MdGRF22* gene were designed by Sangon Biotech (Shanghai, China) Co., Ltd., and the primers are provided in Appendix A. The PCR amplification procedure consisted of 95 °C for 5 min of initial denaturation, 40 cycles of denaturation at 95 °C for 30 s, annealing at 58 °C for 42 s and extension at 72 °C for 1 min, followed by a final extension at 72 °C for 10 min. The amplified products were analyzed by electrophoresis on a 1% agarose gel, and the target bands were recovered. After the pCAMBIA1300 expression vector was ligated, Escherichia coli (*E. coli*) DH5α was transformed using a kit from Tiangen Biotech Co., Ltd. (Beijing, China). The positive colonies were identified by PCR and sequenced by Sangon Biotech (Xi’an) Co., Ltd. (Xi’an, China).

### 4.5. Subcellular Localization of MdGRF22 Protein

The recombinant pCAMBIA1300-*MdGRF22*-GFP and pCAMBIA1300-GFP were transformed into Agrobacterium GV3101 (Beijing Biomed Gene Technology Co., Ltd., Beijing, China), respectively. The resuspended bacterial solution was injected into the abaxial side of *Nicotiana benthamiana* leaves, then cultured in an incubator for two days before being observed using a confocal laser scanning microscope [46].

### 4.6. Y2H Analysis

The recombinant MdGRF22-BD was constructed and transformed into *E. coli* DH5α. The primers for MdGRF22-BD are provided in Appendix A. MdGRF22-BD was transformed into the yeast competent cell AH109, and self-activation and toxicity verification were performed on SD/-Trp and SD/-Trp + X-α-gal plates, respectively.

The apple AD library was co-transformed with BD-MdGRF22 and plated on 50 QDO plates. After four days of culture at 30 °C, white colonies were transferred to the QDO + X-α-gal plates. Blue colonies were picked and cultured in 1 mL QDO liquid medium at 30 °C for two days, followed by PCR amplification using AD universal primers. The sequencing data was compared with the NCBI database to select candidate proteins for further verification. The AD universal primers are provided in Appendix A.

The recombined MdSK-AD was co-transformed with MdGRF22-BD into the yeast competent cell AH109 and plated on DDO and DDO + X-α-gal plates. After two days, blue colonies were transferred to QDO and QDO + X-α-gal plates for culture. The sequences of the primers for MdSK-AD are provided in Appendix A.

### 4.7. Generation of MdGRF22 Transgenic A. thaliana and Apple Callus

The WT *A. thaliana* was of the Columbia ecotype, and the apple callus variety was ‘Orin’, all from the laboratory of College of Horticulture, Gansu Agricultural University. The *A. thaliana* flower buds were shaken in the resuspended bacterial solution (1/2 MS + 5% sucrose + 0.05% Silwet) for 90 s, followed by dark incubation for 24 h. After seven days, the infection was repeated a second time, and the plants were then cultivated until T0 generation seeds were harvested. Concurrently, seeds of the T0 generation of *A. thaliana* were selected on medium containing 50 mg·L^−1^ hygromycin (Hyg) until T3 generation occurred. The transformation of the ‘Orin’ callus was performed according to the method described by Zhang et al. [47]. The apple callus was genetically transformed and screened in MS medium supplemented with 50 mg·L^−1^ Hyg and 300 mg·L^−1^ cefotaxime (Cef).

### 4.8. Identification of the MdGRF22 Transgenic A. thaliana and Callus

A DNA extraction kit (Tiangen Biotech Co., Ltd., Beijing, China) was used to extract DNA from T3 *A. thaliana* leaves and apple callus cultured in a selected medium. PCR amplification was performed with 2 μL of the Hyg upstream and downstream primers, 10 μL of Green Taq Mix enzyme (Nanjing Vazyme Biotechnology Co., Ltd., Nanjing, China), 6 μL of ddH_2_O and 2 μL of DNA. The sequences of the primers for Hyg are provided in Appendix A.

### 4.9. Treatment and Determination of Related Physiological Indexes in A. thaliana and Callus

Two-week-old WT and T3 transgenic *A. thaliana* (OE) were subjected to 15 days of natural drought, followed by a two day rewatering treatment. Each treatment per genotype consisted of three potted plants, with six plants per pot. Callus weighing 0.05 g from the WT and transgenic lines (OE) were placed on MS + 3% sucrose + CH + 1 mg·mL^−1^ 2,4-D + 1 mg·mL^−1^ 6-BA medium, and contained 3% PEG and 6% PEG, respectively [48]. The drought stress lasted for 15 days, with three replicates for each condition.

*A. thaliana* and callus post-PEG treatment were collected, and the physiological indices were determined. Relative electrolyte leakage was measured according to the method described by Jha and Mishra [49]. The activities of MDA, SOD, CAT and POD were assessed using the Suzhou Grace Biotechnology Kit (Jiangsu, China).

### 4.10. Statistical Analysis

Data analysis was conducted using IBM SPSS21, and plotting was performed by Origin 2018. The results of the data analysis were expressed as the mean ± standard error (SE) of three independent experiments. The one-way ANOVA test (*p* < 0.05) was used to analyze the differences between treatments.

## 5. Conclusions

The study reveals that *MdGRF22* and its homologous genes possess similar sequences and structures. The subcellular localization analysis indicates that the MdGRF22 protein is localized in both the cytoplasm and cell membrane. The Y2H assay demonstrates that the MdGRF22 protein may interact with MdSK. Overexpression of *MdGRF22* reduces drought tolerance in transgenic *A. thaliana* and ‘Orin’ calluses. Our study provides a theoretical basis for understanding the response of the apple 14-3-3 family to abiotic stress. However, the regulatory mechanism of the MdGRF22 protein on drought stress is very complex, and further studies at multiple molecular levels are still needed to confirm its function in the future.

## Figures and Tables

**Figure 1 plants-14-01968-f001:**
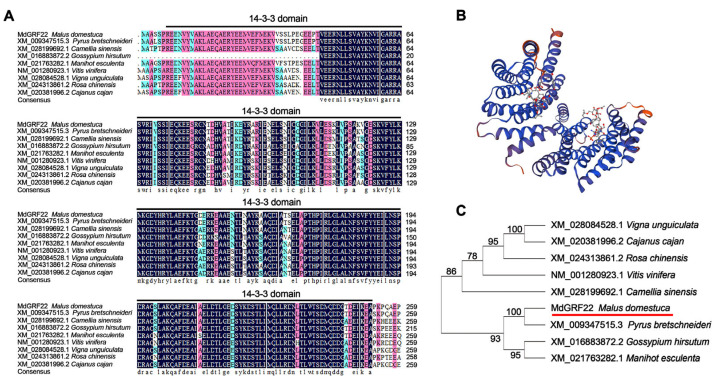
Analysis of sequence characteristics of *MdGRF22* and homologous genes. (**A**) Multiple sequence alignment of *MdGRF22* and homologous genes across nine species. The black, red and blue backgrounds represent amino acid similarities of 100%, over 80% and over 60%, respectively. (**B**) Protein model analysis of the MdGRF22 protein. (**C**) Phylogenetic tree of *MdGRF22* and eight homologous genes. Constructed using the NJ method with 1000 bootstrap replications. The red underline indicates the *MdGRF22* gene.

**Figure 2 plants-14-01968-f002:**
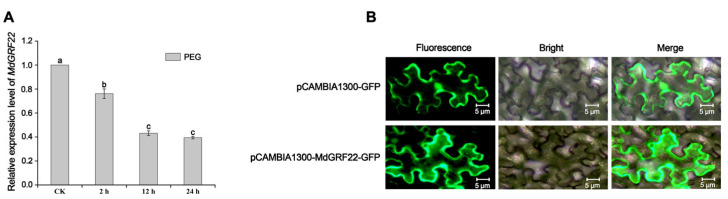
qRT-PCR and subcellular localization analysis of the *MdGRF22* gene. (**A**) Expression levels of *MdGRF22* gene after 0, 2, 12 and 24 h treatment in PEG. (**B**) The pCAMBIA1300-*MdGRF22*-GFP construct was transiently transfected into tobacco as an expression vector, with the pCAMBIA1300-GFP vector serving as a control. The scale bar represents 5 µm. Different lowercase letters indicate a significant difference at *p* < 0.05.

**Figure 3 plants-14-01968-f003:**
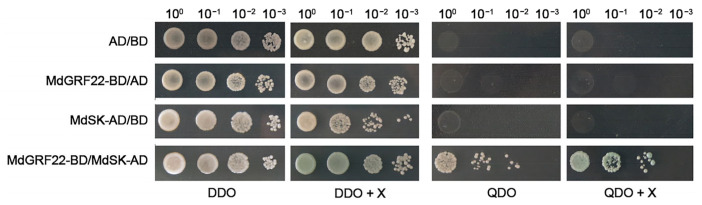
Y2H analysis of MdGRF22 protein. AD and BD represent pGADT7 and pGBKT7, respectively. DDO designates SD/-Leu-Trp, QDO designates SD/-Ade-His-Leu-Trp, and X designates X-α-gal. The numbers 10^0^, 10^−1^, 10^−2^, 10^−3^ represent dilution ratios.

**Figure 4 plants-14-01968-f004:**
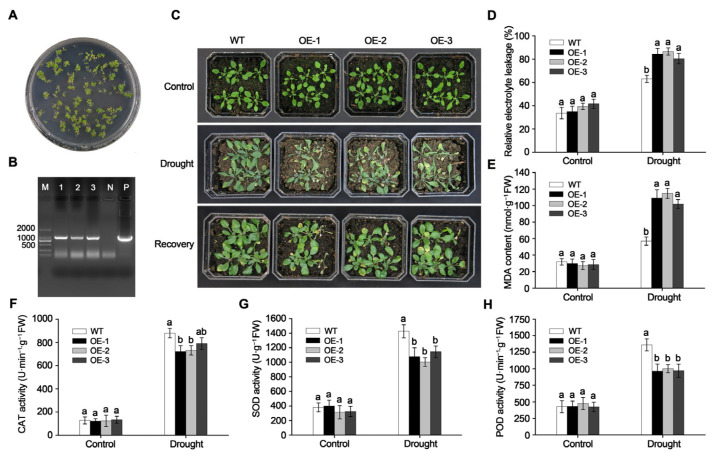
Drought treatment and physiological index determination in *MdGRF22* transgenic *A. thaliana.* (**A**) Screening of *MdGRF22* transgenic *A. thaliana*. (**B**) PCR detection of transgenic *A. thaliana* lines. M represents the DL2000 Marker, and the bands 1, 2 and 3 represent the positive lines. N represents the WT, and P represents the positive control. (**C**) Phenotype responses of *A. thaliana* lines to drought treatment. Control refers to the before treatment, while recovery indicates the state after re-watering following drought stress. (**D**–**H**) Relative electrolyte leakage, MDA content, CAT, SOD and POD activity, respectively. Different lowercase letters indicate a significant difference at *p* < 0.05.

**Figure 5 plants-14-01968-f005:**
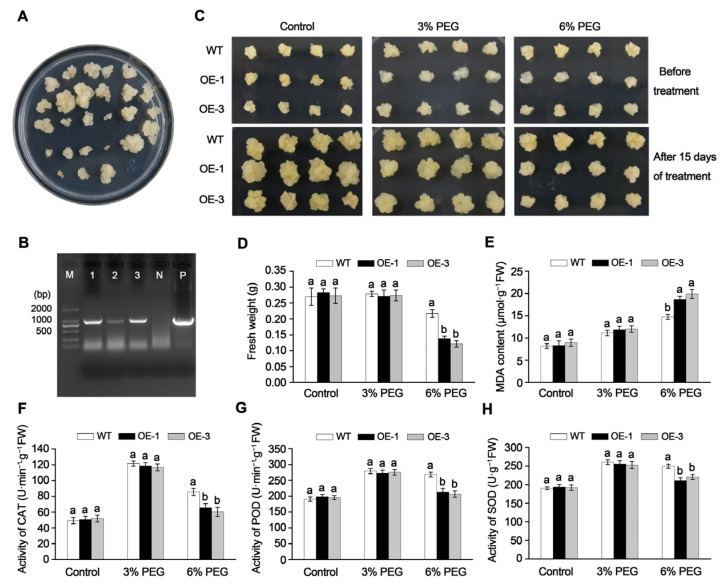
Drought treatment and physiological index determination in *MdGRF22* transgenic callus. (**A**) Screening for transgenic ‘Orin’ callus. (**B**) Identification of *MdGRF22* in transgenic callus. M represents the DL2000 Marker, and the bands 1 and 3 represent the transgenic callus. N represents the WT, and P represents the positive control. (**C**) Phenotypic changes in callus after 15 days of treatment. (**D**) Fresh weight of callus. (**E**–**H**) depict MDA content, relative conductivity, CAT, POD and SOD activity, respectively. Different lowercase letters indicate a significant difference at *p* < 0.05.

**Figure 6 plants-14-01968-f006:**
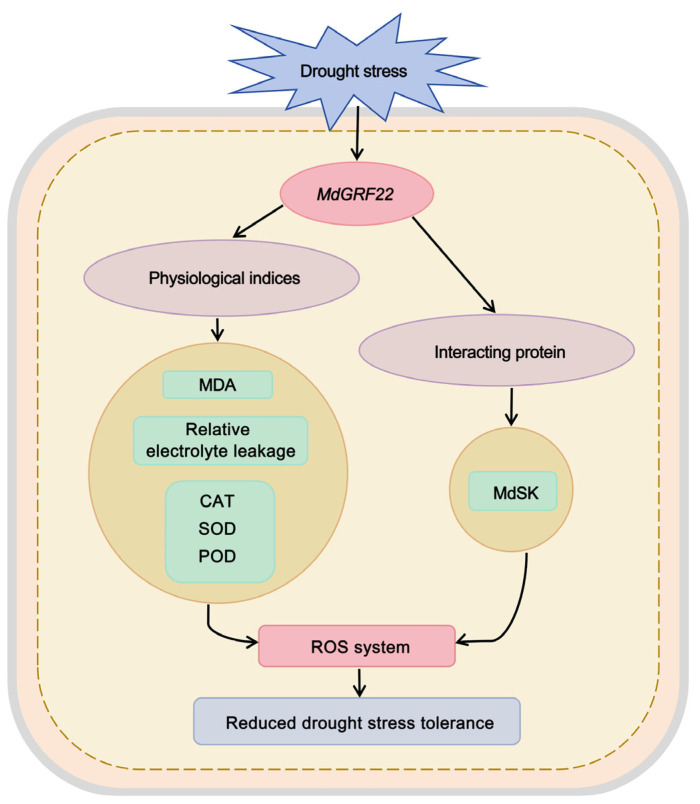
The tolerance model demonstrates that *MdGRF22* negatively regulates drought stress. The black arrow represents the facilitative effect.

## Data Availability

Data are contained within the article and Appendix A.

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
