# Peer review of "MdGRF22, a 14-3-3 Family Gene in Apple, Negatively Regulates Drought Tolerance via Modulation of Antioxidant Activity and Interaction with MdSK"

_plants, 2025, doi:10.3390/plants14131968_

Round 1
Reviewer 1 Report
Comments and Suggestions for Authors
Section 1 needs to specify the aim of this work.
Section 4.1. How did the authors calculate the sample size for this study? What is the "n" for each treatment?
The subcellular localization analysis of the MdGRF22 gene was carried out at 0, 2, 12, and 24 h. Did the authors consider showing the results of this experiment at 6 h?
Was there a taxonomic identification of Royal Gala’ apple seedlings?
Lines 263-271 present a high level of plagiarism.
What are the limitations of this study?
Can the authors provide a hypothesis regarding the mechanisms of the interaction between MdGRF22 and its target proteins? Are there previous reports on this topic?
Why did the authors use the Duncan test? Did the data present normality?
There are 3 references to the corresponding author. Are all the references necessary for this study?
Reviewer 2 Report
Comments and Suggestions for Authors
The manuscript "MdGRF22, a 14-3-3 Family Gene in Apple, Negatively Regulates Drought Tolerance via Modulation of Antioxidant Activity and Interaction with MdSK" reports an interesting research but must be improved to achieve a scientific quality for being publishable.
Due to the scope of the journal, Introduction should start with reference to Malus domestica and challenges by stress in this plant species. Then, references to both proteins under study (MdGRF22 and MsSK) should be introduced. Finally, an explicit general objective and some specific objectives, and not phrases explaining the results, ought to be added. This is especially important because, according to the Instructions for Authors, Introduction is followed by Results. This latter section contains a great variety of experiments, from bioinformatic to transformation activities, and includes tobacco, Arabidopsis, yeast and apple as empirical materials, being difficult to relate with the problem presented in Introduction without specifying the particular objetives to which these experiments account for,
Then, Materials and method sections must be redacted in the same order than Results are presented, and explained as previously suggested their relationship with the objectives.
Once modified these three sections, Discussion and Conclusions must be rewritten for adjusting to the objectives and eventually hypothesis that justified the research.
Round 2
Reviewer 2 Report
Comments and Suggestions for Authors
Authors have modified according to this reviewer's suggestion and the scientific quality of the manuscript has been greatly improved.